# *Enterococcus faecalis*-Aided Fermentation to Facilitate Edible Properties and Bioactive Transformation of Underutilized *Cyathea dregei* Leaves

Israel Sunmola Afolabi [1,*], Aderinsola Jumai Adigun [1], Precious Amaneshi Garuba [1], Eze Frank Ahuekwe [2], Oluwatofunmi E. Odutayo [1] and Alaba Oladipupo Adeyemi [1]

1 Biochemistry Department, College of Science and Technology, Covenant University, Canaanland, Ota 112233, Nigeria; adigun.aderinsola@stu.cu.edu.ng (A.J.A.); amaneshigaruba@gmail.com (P.A.G.); oluwatofunmi.obaseki@gmail.com (O.E.O.); ola.adeyemi@covenantuniversity.edu.ng (A.O.A.)
2 Biological Sciences Department, College of Science and Technology, Covenant University, Canaanland, Ota 112233, Nigeria; eze.ahuekwe@covenantuniversity.edu.ng
* Correspondence: israel.afolabi@covenantuniversity.edu.ng; Tel.: +234-803-392-3264

**Abstract:** *Cyathea dregei* (CD) is a weed plant that is rarely consumed. This study investigated the impact of *Enterococcus faecalis* as an agent of 3–5-day fermentation, thus stimulating the edible properties of the leaves from CD using *Talinum fruticosum* as the control. The proximate content, biochemical, antioxidant properties, and phytochemical constituents of the unfermented and fermented leaves were examined. The lactate dehydrogenase activity (LDH) activity significantly increased ($p < 0.05$) due to the fermentation, which peaked on the third day. The fat, ash, and crude fiber constituents of the fermented CD leaves were significantly higher ($p < 0.05$), especially on day 3, compared to the unfermented leaves of CD. The leaves of CD naturally possess significantly higher ($p < 0.05$) values of calcium, selenium, magnesium potassium, sodium, zinc, and vitamin C but significantly lower ($p < 0.05$) values of vitamins A and E compared to those of water leaf. The fermentation aided the synthesis of caffeic acid (61.71 mg/10 g extract), eleven other bioactive phytochemicals (0.14–60.24 mg/10 g extract), two unexplored saponins (P-Scd, 52.05 mg/10 g extract), and a phenolic compound (P-Pcd, 0.23 mg/10 g extract). Four novel intermediary compounds and six other established compounds were freshly identified with fermentation. The leaves of *C. dregei* are naturally rich in bioactive nutrients and phytochemicals that trigger their strong antioxidant qualities, which were improved by this fermentation technique. *E. faecalis* is most likely to engage LDH in driving the fermentation transforming the *C. dregei* into a potential edible vegetable.

**Keywords:** fermentation; biotransformation; vegetable; food security; bioactive; nutrients; phytochemicals; antioxidants; quality

## 1. Introduction

Fermentation is key to guaranteeing global food insecurity strategies that ensure the availability of food on the table around the world and the extension of its shelf-life. It is most essential in areas where people's purchasing power is insufficient to ensure reliable access to food [1,2]. There is an increasing global demand for a proactive tool to create more natural foods as sources of nutrients that can accommodate the rapidly growing global population. Currently, there are about 800 million people worldwide who lack access to food, and this number is expected to increase to about 1.2 billion people; therefore, about one-fifth of the global population will be affected by food insecurity. Among the billions of people, a sufficient number are deprived of basic micronutrients required by the average human being [3]. The proportion of this global population is increasingly affected by malnutrition caused by poor income. Malnutrition weakens the immune system, psychological morale, mental strength, and body frame and impedes growth in

humans as found in kwashiorkor [4,5]. The phenomenon is associated with human apathy, anxiety, depression, and childhood death due to opportunistic diseases [6,7]. Vegetables are a major instrument for combating undernutrition among the less privilege populace since they are cheap and rich in vital nutrients required for human sustenance. Most of these less privileged populations are individuals who rarely have access to quality foods [8–10].

Humans have successfully employed fermentation to improve the nutritional status and extend the shelf-life of nutrient-endowed foods to sustain global food security. Unique fermented food varieties, such as cheese, milk, beer, wines, and yogurt, have emerged due to these extensive traditional efforts [11]. Such understanding has been successfully translated to produce health-beneficial fermented vegetables such as kimchi and sauerkraut, which are often considered the most well-known fermented vegetables [9]. *Cyathea dregei* is indigenous to Southern Africa and a common tree fern weed popularly known as "gewone boomvaring" in that region. It is categorized as a shrub and perennial plant that is geographically located across the globe, especially in Africa, such as Madagascar, Angola, Ghana, Malawi, Nigeria, Rwanda, Uganda, and Zambia. This plant has dark trunks with huge and bright green fronds. These species grow well under the full sun along swamp edges, open streambanks, grassland, riverine forest, and occasionally in scrub along mountain streams and forests.

Fermented leafy vegetables are advantageous to the human immune system and well-being, and sauerkraut and kimchi are among the most popular fermented vegetables [1,3]. As food insecurity increases, food resources continue to diminish, and it is postulated that numerous underexploited plant leaves could act as substitute sources of nutrients and foods when their positive potential is properly utilized. Fermentation can augment the nutritional value of such plant leaves [1]. Reports on the food processing and nutritional status of the leaves of *C. dregei* are rare. The research aims to assess the influence of *Enterococcus faecalis*-assisted fermentation in re-engineering the edible properties of the leaves from *C. dregei*. This specific fermentation process aided by lactic acid bacteria also focuses on the possibility of turning the plants for edible purposes and influencing the nutritional properties of the plants for the benefit of mankind, and phytochemical transformation is identified as the lactic acid bacteria source for carbon sources.

## 2. Materials and Methods

### 2.1. Chemicals

All reagents and kits used for this study were of analytical grade and procured from established reputable manufacturers as previously detailed [3]. The HPLC grade standards used were mainly from Lichrosolv, Darmstadt, Germany.

### 2.2. Collection of Plant Leaves

Leaves from *C. dregei* were obtained from within the heathland of Cannan City, Ota, Ogun State, Nigeria. The *C. dregei* plant was presented to an expert in the Department of Biological Sciences at Covenant University, who facilitated the identification and the subsequent deposition at the Forest Research Institute of Nigeria, where it was assigned a voucher number (FHI 112796). The edible leaves from *T. fruticosum*, which is commonly called water leaf, and ewe-gbure by the Yoruba people living in Nigeria were adopted as a benchmark for this experimentation study.

### 2.3. Handling of Lactic Acid Bacteria for the Experimentation

The method described by Afolabi, Ahuekwe, Garuba, Adigun, Odutayo, and Adeyemi [3] was followed for the collection, culturing, and identification of the *Enterococcus faecalis* (MW481698) earlier authenticated as Gram-positive, catalase-negative, and non-hemolytic and deposited in the NCBI database [11]. The previously isolated bacteriocin-producing *Enterococcus faecalis* (MW481698) employed in the study was identified as non-hemolytic catalase-negative and Gram-positive bacteria [12]. A loopful of the isolate was suspended in 2 mL MRS broth before inoculation in a BHI broth and incubated at 200 rpm and 37 °C

for 4–6 h in a shaker incubator. Before fermentation, both cultures were propagated twice in the respective media and further maintained in the BHI and MRS media with glycerol (40% *v/v*) at 40 °C.

### 2.4. Fermentation Procedure Setup to Harness the Edible Properties of Plant Leaves

The collected, air-dried plant leaves were further processed before and during fermentation as previously described [3]. The materials were sterilized at 121 °C for 30 min at 15 psi. Inoculation was carried out under aseptic conditions, and the fermentation jars were incubated under controlled conditions of ambient temperature of 37 °C for 3 days (72 h) or 5 days (120 h).

### 2.5. Monitoring of Quality Parameters after Fermentation

The fermented samples of both the control leaves and *C. dregei*-aided fermentation inoculum were homogenized and sieved to obtain their filtrates and the residues for additional scrutiny. The residues were oven-dried (GenLab OV/200/SS/F/DIG oven, Widnes WA8 0SR, UK) and examined for bulk nutrients and mineral composition, while the liquid portion (filtrate) drained from these leaves was apportioned into two parts. The first filtrate (FA) was analyzed for the determination of the level of hydrogen ion concentration (pH), proximate components, antioxidant qualities (FRAP, DPPH, and total antioxidant capacities), lactic acid dehydrogenase activity, and estimation of the flavonoids, phenols, and saponin groups, and the unique chemical composition determined using the GC-MS method. The other part of the filtrate (FB) was concentrated to a greater extent using a rotary evaporator at 20 °C for periods of 30 and 20 min for water leaf and *C. dregei*, respectively. The resultant concentrated filtrates (CFs) were restored before using high-performance liquid chromatography (HPLC) to identify and quantify the individual compounds within the phenol, flavonoid, and saponin groups and the antioxidant vitamins (A, C, and E).

#### 2.5.1. Quantitative Chemical and Phytochemical Constituent Determination

The method used to determine the level of hydrogen ion concentration (pH) was as previously described [3]. The proximate composition was determined using the AOAC method [13]. The total protein concentration was investigated spectrometrically (Thermo Scientific Ltd. model: GEN10S UV-Vis, serial no: 2L5V095205, Waltham, MA, USA) using the biuret method [14] and the total carbohydrate content of the extracts using the Anthrone reagent method [15]. Both quantification procedures were carried out at 540 nm and 620 nm wavelengths, respectively. The saponin, flavonoid, and total phenol constituents were determined quantitatively according to the method described by Olawole et al. [16].

#### 2.5.2. Assay for Antioxidant Qualities of the Extracts

The individual antioxidant qualities indicators were examined spectrophotometrically (Thermo Scientific Ltd., model: GEN10S UV-Vis, serial no: 2L5V095205) in the extracts from the fermented leaves from *C. dregei* and control leaves (water leaf). The absorbance values reflected at 700 nm and 695 nm wavelengths were explored to determine the corresponding ferric-reducing antioxidant power (FRAP) and total antioxidant capacity (TAC), respectively, following standard procedure [17]. The abilities (% inhibition) of the extracts to scavenge diphenyl-1-picrylhydrazyl radical scavenging activity (DPPH) radicals were determined at 517 nm wavelength, calculated using the following stated formula, and compared with that expressed by vitamin C standard [18].

$$\% \text{ Inhibition} = \frac{\text{Optical density (control)} - \text{Optical density (sample)}}{\text{Optical density (control)}} \times 100$$

#### 2.5.3. Procedure for Preparing the Sample Leaves and Micronutrient Analysis Using Atomic Absorption Spectrometric Technique

The ashed samples (1.0 g) from the leaves were digested using the AOAC wet-ash method to digest the ashes from the leaf samples [19]. The level of each of the six mi-

cronutrients (sodium, calcium, selenium, magnesium, potassium, and zinc) considered in the digest was thereafter ascertained using an atomic absorption spectrophotometer (Shimadzu, model AA-7000, Nishi-ku, Japan) as previously described by Afolabi et al. [20], while atomic absorption spectrometry (Buck Scientific 210VGP, East Norwalk, CT, USA) was similarly utilized to determine the heavy metals (copper, iron, chromium, cadmium, nickel, and lead) as previously described [3].

### 2.5.4. HPLC Procedure for the Determination of Vitamins in the Sample Leaves

Both categories of vitamins were performed using the Agilent 1200-HPLC procedure that was coupled with a diode array (DA) detector as detailed by Afolabi, Ahuekwe, Garuba, Adigun, Odutayo, and Adeyemi [3]. An aliquot (10 µL) of the extracted sample that was previously mixed with the respective mobile phase (4.0 mL) was analyzed using a 4.6 nm × 150 nm, 5 µL size Zorbax Eclipse XDB-C18 type of column at 40 °C for both the fat-soluble A and E vitamins of interest and the water-soluble vitamin C, including their standards. A mixture of acidified water (pH 2.16) and acetonitrile (99:1 *v/v*) pumped at a flow rate of 2.0 mL/min was used as a mobile phase for vitamin C analysis, while undiluted absolute methanol pumped at a flow rate of 2.0 mL/min served as the mobile phase for the assay for the water-soluble vitamins of interest. Concentrations of 1.08, 1.0, and 1.0 mg/mL were used for the vitamin A, E, and C standards used during the analysis, respectively. The concomitant chromatograms for this assay are as indicated (Figures S1–S11).

### 2.5.5. Preparation and HPLC Procedure for Quantitative Determination of Individual Phytochemical Constituents of the Sample Leaves

An aliquot of 10 mL of methanol solution (70% *v/v*) was dispensed in the extract (0.1 g) from the leaves, agitated, and left to stand for 1–2 h in a reaction container. The mixed samples were centrifuged with a refrigerated centrifuge (model: CR21G, serial no: S2025709), and the supernatant was collected and drained through a micron filter into another sample container. The individual compound that constitutes the bioflavonoid, saponins, and phenolic fractions of the filtrate obtained from the leaves was separated, identified, and quantified using prescribed HPLC methods [3]. Mixtures of acetonitrile/water/formic acid, acetonitrile/water, and acetonitrile/water/acetic acid mixed at a ratio of 25:74:1, 70:30, and 19:80:1 served as the solvents for the bioflavonoid, saponin, and phenolic analysis, respectively. A detector (LC-8518 diode array) set at wavelengths of 210, 205, and 272 nm, a column (150 mm × 4.6 mm) set at 40, 40, and 35 °C, and runtimes of 25, 14, and 25 mins were employed for the analysis of the 40 µL bioflavonoid, saponin, and phenolic samples, respectively.

The detailed volatile phytochemicals in the extracts were identified as previously described [21]. An optima 5 MS capillary column (30 × 0.25 mm) conditioned at a temperature of 60 °C and a flow rate of 3.22 mL/min formed the GC-MS (Shimadzu QP2010SE, Osaka, Japan) conditions along with 144.4 kPa pressure and programmed oven temperatures of 60, 120, and 290 °C, with running times of 2, 2, and 3 min, respectively, for the analysis of the volatile compounds in the 1.0 µL aqueous extracts. The resultant chromatograms for the phytochemical compounds (Figures S12–S20) and those for the volatile compounds (Figures S21–S23) were as indicated. The compounds identified from both the HPLC and GC-MS analysis in this study were sequenced and integrated into a proposed scheme to elucidate the biochemical processes using MarvinSketch software (15.9.14.0, ChemAxon Limited, Budapest, Hungary). Identified volatile compounds from GC-MS analysis with less than 60% similarity index were excluded from further consideration in the postulation of the scheme.

### 2.5.6. Procedure for Lactate Dehydrogenase Activity Analysis

The semi-micro method prescribed in the manual that accompanied the procured kit (Randox, Crumlin, Northern Ireland) was employed to assess the lactic acid dehydrogenase (LDH) activity using the specified semi-micro precision method spectrophotometrically at 37 °C and 340 nm wavelength. A portion of the liquid sample (0.02 mL) derived from the leaves was pipetted into a cuvette containing 1 mL of a reagent consisting of nicotinamide adenine dinucleotide (NADH) and buffered substrate (pH 7.5). The absorbances were determined at 0 s and intervals of 60 s for 180 s immediately after mixing.

The resultant enzyme activities (U/l) were calculated using the following formula.

$$U/l = 8095 \times \Delta \text{ Absorbance (340 nm)/min}$$

### 2.6. Method of Statistical Analysis

All experimental procedures were conducted in triplicate, and the results were reported as mean ± standard deviation (SD) and evaluated using one-way analysis of variance (ANOVA) in MegaStats software (version 10.3 Release 3.2.1). The results were considered significant at $p < 0.05$.

### 3. Results

#### 3.1. Effect of Fermentation on the Acidity of the Leaves of C. dregei

The pH of the *C. dregei* leaves significantly increased ($p < 0.05$) in a time-dependent manner (Table 1). The acidity of the fermenting medium reduced as expected, from 3.98 to 9.0, as the fermentation proceeded (Table 1), indicating an increase in alkalinity during the fermentation. This is unlike the expected increased acidity that is typical of fermentation. This pattern change in the ionic content of the fermenting medium suggests the fermentation process was encouraged by the alkaline environment. The profile of the proximate composition of unfermented and fermented leaves from *C. dregei* and *T. fruticosum* was also indicated (Table 1). The carbohydrate content in the leaves of *C. dregei* was significantly higher ($p < 0.05$) than that of *T. fruticosum*. The carbohydrate values were significantly decreased ($p < 0.05$). In general, the fermentation significantly increased ($p < 0.05$) the carbohydrate content of *T. fruticosum*. The protein content in the leaves of *C. dregei* was significantly higher ($p < 0.05$) than that of *T. fruticosum*. The protein content values were significantly increased ($p < 0.05$) for *C. dregei* and *T. fruticosum* on day 3 by the fermentation process. In general, the fermentation significantly lowered ($p < 0.05$) the carbohydrate content of *T. fruticosum*. The crude fiber content in the leaves of *C. dregei* was significantly lower ($p < 0.05$) than that of *T. fruticosum*. The crude fiber values were significantly increased ($p < 0.05$), mainly for *C. dregei* on day 3 by the fermentation process. In general, the fermentation significantly lowered ($p < 0.05$) the crude fiber content for *T. fruticosum*. The fermentation significantly increased ($p < 0.05$) the fats and ash contents of *C. dregei* leaves (Table 1). Fermentation significantly reduced the moisture level in the *C. dregei* leaves.

#### 3.2. The Nutritional Attributes of the Fermented Leaves of C. dregei

The leaves of *C. dregei* naturally possess significantly higher ($p < 0.05$) values of calcium, selenium, magnesium potassium, sodium, zinc, and vitamin C, indicating that the leaves possess superior micronutrients compared to the water leaf, a commonly consumed vegetable leaf. The leaves of *C. dregei* were only found to have significantly lower ($p < 0.05$) values of vitamins A and E relative to the values of the water leaf (Table 2). The amounts of minerals in the CD leaf were identical to those of the water leaves that represent the commonly consumed vegetables. The CD leaves were richer in the important basic microminerals such as sodium, potassium, calcium, magnesium, and selenium. The underutilized leaves from *C. dregei* were rich in the necessary and relatively stable mineral nutrients required for the sustenance of consumers, suggesting they were not impeded by the fermentation induced by probiotic lactic acid bacteria (*E. faecalis*). The leaves and the

fermented leaves of *C. dregei* were rich in mineral iron, with the potential to easily provide the recommended daily intake for the mineral. The plant may be essential for enriching blood and treating anemia, sickle cell disease, and other anemia-related diseases. These leaves from *C. dregei* were also free of highly toxic elemental lead, indicating that there may be no heavy metal toxicity threat associated with their consumption. Notably, there was a gradual depletion of the sodium content of *C. dregei* in a time-dependent manner during the fermentation, suggesting it was a required substrate to facilitate the active biochemical process of *E. faecalis*-induced fermentation (Table 2). The fermentation significantly reduced ($p < 0.05$) the heavy metals (copper, iron, manganese, and cadmium) considered in this study.

**Table 1.** pH and bulk nutrients in the fermented *C. dregei* Leaves.

| Parameters | Leaf Types | Duration of Fermentation (Days) | | |
|---|---|---|---|---|
| | | 0 | 3 | 5 |
| pH | WL † | 4.31 ± 0.02 [a] | 7.50 ± 0.00 [b] | 8.00 ± 0.00 [c] |
| | CD | 3.98 ± 0.04 [a] | 8.49 ± 0.01 [b] | 9.00 ± 0.00 [c] |
| Moisture (%) | WL † | 11.62 ± 0.03 [a] | 20.77 ± 0.06 [b] | 23.49 ± 0.00 [c] |
| | CD | 16.32 ± 0.00 [a] | 10.39 ± 0.01 [b] | 10.69 ± 0.00 [b] |
| Protein (%) | WL † | 12.20 ± 0.10 [a] | 21.40 ± 0.10 [b] | 12.13 ± 0.06 [c] |
| | CD | 14.23 ± 0.05 [a] | 24.07 ± 0.05 [b] | 15.43 ± 0.05 [b] |
| Fats (%) | WL † | 14.34 ± 0.11 [a] | 19.10 ± 0.17 [b] | 29.90 ± 0.09 [c] |
| | CD | 2.57 ± 0.05 [a] | 1.78 ± 0.10 [b] | 20.60 ± 0.04 [c] |
| Carbohydrates (%) | WL † | 17.54 ± 0.04 [a] | 18.36 ± 0.13 [b] | 16.98 ± 0.04 [c] |
| | CD | 31.02 ± 0.09 [a] | 18.12 ± 0.06 [b] | 12.41 ± 0.09 [c] |
| Crude fiber (%) | WL † | 32.56 ± 0.01 [a] | 10.01 ± 0.01 [b] | 11.22 ± 0.01 [c] |
| | CD | 15.54 ± 0.00 [a] | 19.01 ± 0.00 [b] | 11.22 ± 0.00 [c] |
| Ash (%) | WL † | 11.73 ± 0.01 [a] | 10.35 ± 0.01 [b] | 6.27 ± 0.01 [c] |
| | CD | 20.30 ± 0.00 [a] | 27.11 ± 0.00 [b] | 25.42 ± 0.00 [c] |

Value = mean ± standard deviation (*n* = 3); [a–c] data within each row with different superscript letters are statistically significantly different ($p < 0.05$). Key: CD = *C. dregei*; WL = water leaf (*T. fruticosum*). Source †: Afolabi, Ahuekwe, Garuba, Adigun, Odutayo, and Adeyemi [3].

From the antioxidant vitamin perspective, the CD was depleted in the values of vitamins A and E and more abundant in vitamin C relative to their corresponding values in the water leaf, representing the edible leafy vegetables. The fermentation diminished the amounts of vitamins A, C, and E in the *C. dregei* leaves, suggesting the preference of the *E. faecalis* fermenting organism for these vitamins during the fermentation (Table 2).

### 3.3. Antioxidant Attributes in the Fermented Leaves of C. dregei

The leaves of *C. dregei* and the commonly consumed vegetable (water leaves) in their natural state possess significantly lower ($p < 0.05$) ability to scavenge disease-causing free radicals compared to the vitamin C standard (Table 3). The fact that *E. faecalis*-induced fermentation significantly increased ($p < 0.05$) the free-radical-scavenging potential of both leaves beyond the capacity of vitamin C is commendable (Table 3). It is therefore possible that a strong antioxidant compound(s) was synthesized by the organism, especially during the 5-day fermentation (Table 3). The levels of FRAP were similarly significantly increased ($p < 0.05$) in antioxidant capacity during fermentation, but the reverse was the case concerning the antioxidant capacity during the fermentation of water leaf. TAC was similarly significantly increased ($p < 0.05$) in *C dregei* and water leaf after 3 and 5 days of *E. faecalis*-induced fermentation, respectively (Table 3).

**Table 2.** Some microminerals in fermented *C. dregei* leaves.

| Mineral | Leaf Type | Duration of Fermentation (Days) | | | Recommended Daily Intake [†] |
|---|---|---|---|---|---|
| | | 0 | 3 | 5 | |
| Ca (mg/L) | WL [¥] | 0.51 ± 0.00 | NA | NA | 1000 mg |
| | CD | 0.81 ± 0.01 [a] | 0.87 ± 0.02 [b] | 0.85 ± 0.02 [c] | |
| Se (mg/L) | WL [¥] | 0.23 ± 0.00 | NA | NA | 25–34 µg |
| | CD | 0.28 ± 0.02 [a] | 0.25 ± 0.00 [b] | 0.23 ± 0.00 [c] | |
| Mg (mg/L) | WL [¥] | 0.69 ± 0.00 | NA | NA | 400 mg |
| | CD | 0.87 ± 0.02 [a] | 0.84 ± 0.00 [b] | 0.83 ± 0.02 [c] | |
| K (mg/L) | WL [¥] | 1.93 ± 0.00 | NA | NA | 3500 mg |
| | CD | 2.18 ± 0.02 [a] | 2.31 ± 0.02 [a] | 2.26 ± 0.02 [a] | |
| Na (mg/L) | WL [¥] | 0.49 ± 0.00 | NA | NA | 2400 mg |
| | CD | 0.57 ± 0.00 [a] | 0.54 ± 0.00 [b] | 0.51 ± 0.01 [c] | |
| Zn (mg/L) | WL [¥] | 0.80 ± 0.00 | NA | NA | 15 mg |
| | CD | 0.84 ± 0.00 [a] | 0.91 ± 0.01 [b] | 0.87 ± 0.01 [c] | |
| Cu (mg/L) | WL [¥] | 0.19 ± 0.00 [a] | 0.28 ± 0.16 | 0.00 ± 0.00 [b] | 2 mg |
| | CD | 0.30 ± 0.00 [a] | 0.25 ± 0.00 [a] | 0.23 ± 0.00 [b] | |
| Fe (mg/L) | WL [¥] | 8.73 ± 0.07 [a] | 16.52 ± 0.11 [b] | 0.00 ± 0.00 [c] | 18 mg |
| | CD | 18.75 ± 0.00 [a] | 11.91 ± 0.00 [b] | 14.94 ± 0.00 [c] | |
| Cd (mg/L) | WL [¥] | 0.05 ± 0.00 [a] | 0.05 ± 0.00 | 0.00 ± 0.00 [b] | 3.6 µg/kg bw |
| | CD | 0.07 ± 0.03 [a] | 0.05 ± 0.00 [b] | 0.05 ± 0.00 [b] | |
| Mn (mg/L) | WL [¥] | 1.24 ± 0.00 [a] | 0.91 ± 0.01 | 0.00 ± 0.00 [b] | 2 mg |
| | CD | 4.00 ± 0.00 [a] | 2.18 ± 0.00 [b] | 2.54 ± 0.00 [b] | |
| Pb (mg/L) | WL [¥] | 0.00 ± 0.00 | 0.00 ± 0.00 | 0.00 ± 0.00 | 1.0 µg/kg bw |
| | CD | 0.00 ± 0.00 | 0.00 ± 0.00 | 0.00 ± 0.00 | |
| Vit. A (mg/mL) | WL [¥] | 0.13 ± 0.00 | NA | NA | 5000 (I.U.) |
| | CD | 0.08 ± 0.00 [a] | 0.05 ± 0.01 [b] | 0.06 ± 0.01 [c] | |
| Vit. C (mg/mL) | WL [¥] | 0.61 ± 0.00 | NA | NA | 60 mg |
| | CD | 2.12 ± 0.00 [a] | 1.85 ± 0.06 [b] | 0.20 ± 0.01 [c] | |
| Vit. E (mg/mL) | WL [¥] | 0.82 ± 0.00 | NA | NA | 30 (I.U.) |
| | CD | 0.57 ± 0.00 [a] | 0.33 ± 0.01 [b] | 0.03 ± 0.00 [c] | |

Value = mean ± standard deviation ($n$ = 3); [a–c] data within each row with different superscript letters are statistically significantly different ($p < 0.05$). Key: NA = not applicable; I.U. = international unit; CD = *C. dregei*; WL = water leaf (*T. fruticosum*). [¥] Sources: Afolabi, Ahuekwe, Garuba, Adigun, Odutayo, and Adeyemi [3]; [†] Source: Afolabi, Nwachukwu, Ezeoke, Woke, Adegbite, Olawole, and Martins [20].

### 3.4. Antioxidant Attributes of the Fermented C. dregei Leaves

The saponin, phenols, and flavonoid levels increased significantly ($p < 0.05$) in both the leaves of *C. dregei* and water leaf throughout their 5-day *E. faecalis*-induced fermentation (Table 3). Thus, the general boost in the antioxidant strength of these leaves could be attributed to the synthesis of compounds in these three groups by *E. faecalis*.

**Table 3.** Phytochemical constituents and antioxidant attributes in the fermented leaves of *C. dregei*.

| Days | Phenol (mgGAE/g) $\times 10^{-2}$ | | Saponin (mgGAE/g) | | Flavonoids (mgGAE/g) $\times 10^{-1}$ | | TAC (µg/mL) | | FRAP (µg/mL) $\times 10^{-5}$ | | | DPPH Inhibition (%) | |
|---|---|---|---|---|---|---|---|---|---|---|---|---|---|
| | WL [¥] | CD | WL [¥] | CD | WL [¥] | CD | WL [¥] | CD | WL [¥] | CD | Vit. C | WL [¥] | CD |
| 0 | 1.88 ± 0.01 [a] | 1.70 ± 0.01 [a] | 1.66 ± 0.08 [a] | 2.93 ± 0.25 [a] | 0.49 ± 0.00 [a] | 1.21 ± 0.00 [a] | 26.38 ± 0.01 [a] | 27.01 ± 0.01 [a] | 24.10 ± 0.00 [a] | 5.32 ± 0.02 [a] | 95.58 ± 0.00 [a] | 75.92 ± 0.25 [a] | 69.29 ± 0.25 [a] |
| 3 | 4.73 ± 0.01 [b] | 6.92 ± 0.01 [b] | 0.90 ± 0.02 [b] | 2.30 ± 0.01 [b] | 0.56 ± 0.01 [b] | 1.42 ± 0.00 [b] | 25.89 ± 0.02 [b] | 27.15 ± 0.00 [b] | 8.49 ± 0.00 [b] | 8.52 ± 0.00 [b] | NA | 70.52 ± 0.00 [b] | 88.86 ± 0.14 [b] |
| 5 | 6.90 ± 0.00 [c] | 8.20 ± 0.01 [c] | 3.23 ± 0.01 [b] | 3.55 ± 0.01 [b] | 0.59 ± 0.01 [c] | 1.56 ± 0.03 [c] | 26.51 ± 0.02 [c] | 26.63 ± 0.02 [c] | 6.72 ± 0.01 [c] | 8.55 ± 0.00 [c] | NA | 80.02 ± 0.14 [c] | 94.60 ± 0.25 [c] |

Results = mean ± SD (*n* = 3). [a–c] The results within the same column with different superscript letters are significantly different ($p < 0.05$). Key: CD = *C. dregei*; WL = water leaf (*T. fruticosum*); NA = Not applicable. [¥] Sources: Afolabi, Ahuekwe, Garuba, Adigun, Odutayo, and Adeyemi [3].

### 3.5. Qualitative Screening of the Phytochemicals Present in the Fermented Leaves of C. dregei

The levels of individual saponin, phenols, and flavonoids detected in *C. dregei* leaves metabolized by *E. faecalis*-assisted fermentation are indicated (Table 4). Thirty compounds, of which five, twelve, and thirteen were saponin, phenolic, and bioflavonoid compounds, respectively (Table 4). The saponins contents in *C. dregei* leaves metabolized by *E. faecalis*-assisted fermentation are indicated (Table 4). The predominant saponin compound detected in the leaves, hederagenin, was completely catabolized by *E. faecalis*, indicating the high preference of the fermenting organism for the compound. The *E. faecalis* in these fermentation environments synthesized furostanol and a new possibly uncharacterized novel saponin compound (S-Scd1) on the third day of fermentation. Extending the fermentation period to 5 days produced two entirely new compounds (spirostanol and aescin). Interestingly, one of these two saponins, spirostanol, was the product of fermenting another plant (*S. monostachyus*) with *E. faecalis* for the same duration. Exploring the phenolic compound transformation in this fermentation, three economically important phenolic compounds, gallic acid, caffeic acid, and ellagic acid, were naturally present in the leaves of *C. dregei*. Caffeic acid and ellagic acid were completely catabolized by the organism after 3 and 5 days of fermentation, respectively, while gallic acid was gradually catabolized at the end of the 5-day fermentation. *E. faecalis*-induced fermentation involved the synthesis of ferulic acid, chlorogenic acid, kaempferol, and chrysin as new phenolic compounds after 3 days of fermentation. Only chrysin of these newly synthesized compounds was completely catabolized at the end of the 5-day fermentation (Table 4).

Furthermore, ascorbic acid was observed as the predominant bioflavonoid compound that naturally occurs in the leaves of *C. dregei*. It is noteworthy that other important antioxidant-bioactive bioflavonoids (chlorogenic acid, gallic acid, quercetin, resorcinol, catechin, and 2,5-dihydroxybenzoic acid) were also naturally present in small quantities (Table 4). The predominant ascorbic acid, along with the gallic acid, quercetin, and resorcinol, were completely catabolized throughout the 5-day *E. faecalis*-induced fermentation. Caffeic acid was increasingly synthesized as the 5-day fermentation progressed. Vanillic acid synthesis at 3 days of fermentation was sustained after 5 days, while chlorogenic acid concentration remained relatively stable throughout the 5-day fermentation.

This study suggests that the types and the number of phytochemical compounds synthesized are functions of the original compound in the plants used for fermentation and the length of exposure of the organisms to the fermenting process. One unidentified phenolic compound (S-Pcd) and one unidentified bioflavonoid compound (S-Bcd) that naturally occur in the leaves of *C. dregei* and another unidentified newly synthesized phenolic compound (P-Pcd) were identified as possible novel compounds during this *E. faecalis*-induced fermentation process. Further efforts are required to validate the authenticity of their novelty and the usefulness of these four unidentified compounds (Table 4).

The compounds identified with GC/MS and their separate concentrations in the leaves of *C. dregei* were as indicated (Table 5). Aescin was identified as a product of fermenting the leaves of *C. dregei*. Vanillic acid, salicylic acid, ellagic acid, coumarin, quercetin, catechin, and resorcinol were part of the important phytochemicals in the *C. dregei* fermentation study (Table 4). Fermentation aided the destruction of resorcinol, which is present in the raw leaves of *C. dregei*. Malonic acid (4.53–4.94%) is a major fatty acid produced, while dimethylethylene glycol (13.68–18.96%) appears to be the major alcohol generated by the 3–5-day fermentation of the leaves of *C. dregei* (Tables 4 and 5). The high level of dimethylethylene glycol compared to the level of malonic acid produced may be the major factor influencing the continual increase in the alkaline level (3.98–8.50–8.90) of the fermenting medium in this study (Table 1). The compound (I4-AeSC) is novel and uniquely implicated in this fermentation study (Figure 1). This study further reveals ascorbic acid as a precursor of chlorogenic acid biosynthesis in plants, which explains why both compounds are often present together in plant foods.

**Table 4.** Phytochemical compounds identified during *E. faecalis*-induced fermentation of *C. dregei* leaves.

| S/N | Identified Compound (Tr (min)[Peak nos.]) | Concentration (mg/10 g Extract) | | |
|---|---|---|---|---|
| | Saponin | Control (Day 0) | 3-Day-Fermented | 5-Day-Fermented |
| 1 | Spirostanol (1.465[8]–1.515[4]) | - | 0.01 | 60.24 |
| 2 | Furostanol (2.382[8]) | - | 47.86 | - |
| 3 | Aescin (2.023[5]) | - | - | 39.72 |
| 4 | P-Scd (3.007[10]) | - | 52.05 | - |
| 5 | Hederagenin (4.373[20]) | 99.77 | - | - |
| | Phenolic | Control (Day 0) | 3-Day-Fermented | 5-Day-Fermented |
| 6 | Caffeic acid (1.198[2]) | 21.75 | - | - |
| 7 | Gallic acid (1.340[3]–1.448[2]–1.573[2]) | 44.83 | 29.81 | 10.44 |
| 8 | Ferulic acid (1.632[3]–1.798[3]) | - | 56.13 | 48.35 |
| 9 | Ellagic acid (2.232[4]–2.532[4]) | 19.11 | 6.04 | - |
| 10 | Chlorogenic acid (2.932[5]–3.073[4]) | - | 5.04 | 11.58 |
| 11 | S-Pcd (3.323[5]–3.165[6]) | 3.52 | 0.09 | - |
| 12 | Luteolin (3.373[6]–3.382[5]) | 3.97 | - | 20.54 |
| 13 | Naringenin (4.007[7]) | 6.74 | - | - |
| 14 | Apigenin (4.990[8]–5.198[7]) | 0.03 | - | 5.33 |
| 15 | Kaempferol (4.998[7]–5.048[6]) | - | 2.74 | 3.53 |
| 16 | Chrysin (6.365[8]) | - | 0.14 | - |
| 17 | P-Pcd (6.573[8]) | - | - | 0.23 |
| | Bioflavonoid | Control (Day 0) | 3-Day-Fermented | 5-Day-Fermented |
| 18 | 2,5-dihydroxybenzoic acid (0.315[2]–0.340[2]) | 0.26 | 0.31 | - |
| 19 | Ascorbic Acid (1.390[3]) | 96.26 | - | - |
| 20 | Caffeic acid (1.498[5]–1.423[1]) | - | 15.19 | 61.71 |
| 21 | O-Coumaric acid (1.565[6]) | - | 2.68 | - |
| 22 | Salicylic acid (1.623[7]) | - | 19.48 | - |
| 23 | Vanillic acid (2.090[8]–2.482[2]) | - | 44.60 | 11.65 |
| 24 | Chlorogenic acid (4.148[5]–4.240[10]–4.257[4]) | 1.27 | 12.58 | 0.18 |
| 25 | S-Bcd (4.323[6]–4.182[9]) | 0.12 | 5.08 | - |
| 26 | Gallic acid (4.448[7]) | 0.65 | - | - |
| 27 | Quercetin (4.948[8]) | 0.87 | - | - |
| 28 | Resorcinol (5.632[10]) | 0.21 | - | - |
| 29 | Catechin (6.815[13]–6.640[12]) | 0.01 | 0.02 | - |
| 30 | Apigenin (2.532[3]) | - | - | 26.47 |

Key: S-Scd = unidentified natural saponin; S-Pcd = unidentified natural phenolic compound; S-Bcd = unidentified natural bioflavonoid compound; P-Scd = unidentified newly synthesized saponins product; P-Pcd = unidentified newly synthesized phenolic product; P-Bcd = unidentified newly synthesized bioflavonoid product; CD = *C. dregei*; WL = water leaf (*T. fruticosum*).

**Table 5.** Compounds identified by GC-MS in fermented leaf extracts of *C. dregei*.

| S/N | Peak | Tr | Area (%) | Similarity Index (%) | Class of Compound | IUPAC Name | Common Name |
|---|---|---|---|---|---|---|---|
| | | | | | Unfermented—Day 0 | | |
| 1 | 1 | 4.017 | 15.05 | 55 | Fluoro-amino acids | 4,4,4-Trifluorothreonine | 4,4,4-Trifluorothreonine |
| 2 | 2 | 6.059 | 2.52 | 89 | Secondary alcohols | 2-Pentanol, 4-methyl- | Isobutylmethylmethanol |
| 3 | 3 | 12.492 | 22.65 | 75 | Hydroxycoumarins | 2H-1-Benzopyran-2-one, 3,4-dihydro- | Hydrocoumarin |
| 4 | 4 | 13.666 | 53.34 | 77 | Terpene ether | 1,7,7-Trimethylbicyclo[2.2.1]heptane-2,5-diol | (1S,4S)-Bornane-2alpha,5beta-diol |
| 5 | 5 | 15.032 | 5.44 | 64 | Fatty acid esters | Ethyl (5E,8E,11E,14E,17E)-5,8,11,14,17-icosapentaenoate | Ethyl 5,8,11,14,17-icosapentaenoate |
| | | | | | Fermented—Day 3 | | |
| 1 | 1 | 4.033 | 4.4 | 62 | Carboxylic acid | (E)-But-2-enyl isobutyl carbonate | Carbonic acid 2-butenyl=ethyl |
| 2 | 2 | 5.213 | 4.53 | 81 | Fatty acid | Propanedioic acid | Malonic acid |
| 3 | 3 | 5.542 | 13.68 | 96 | Alcohol | 2,3-Butanediol | Dimethylethylene glycol |
| 4 | 4 | 11.375 | 11.30 | 90 | Hydroxycoumarins | 2H-1-Benzopyran-2-one, 3,4-dihydro- | Hydrocoumarin |
| 5 | 5 | 12.224 | 63.28 | 93 | Benzopyrones | o-Hydroxycinnamic acid lactone | Coumarin |
| 6 | 6 | 14.996 | 1.21 | 52 | Heterocyclic | N-Phenyl-2-cyclohexene-1-carboxamide | 2-Cyclohexenecarboxanilide |
| 7 | 7 | 15.479 | 1.62 | 60 | Dihydrocarvone | 5-Isopropenyl-2-methyl-2-cyclohexen-1-yl pivalate | Limonen-6-ol, pivalate |
| | | | | | Fermented—Day 5 | | |
| 1 | 1 | 4.025 | 1.42 | 83 | Carboxylic acid | 1,3,4-Trihydroxy-5-oxocyclohexanecarboxylic acid | Cyclohexan-1,4,5-triol-3-one-1-carboxylic acid |
| 2 | 2 | 4.043 | 0.54 | 83 | Carboxylic acid | 1,3,4-Trihydroxy-5-oxocyclohexanecarboxylic acid | Cyclohexan-1,4,5-triol-3-one-1-carboxylic acid |
| 3 | 3 | 5.076 | 4.94 | 79 | Fatty acid | Propanedioic acid | Malonic acid |
| 4 | 4 | 5.498 | 9.30 | 89 | Secondary alcohols | 1,3-Butanediol | Methyltrimethylene glycol |
| 5 | 5 | 11.333 | 18.96 | 87 | Hydroxycoumarins | 2H-1-Benzopyran-2-one, 3,4-dihydro- | Hydrocoumarin |
| 6 | 6 | 12.235 | 64.13 | 93 | Benzopyrones | o-Hydroxycinnamic acid lactone | Coumarin |
| 7 | 7 | 14.999 | 0.70 | 61 | Sulfated steroids | 5.alpha.-Pregnan-3.beta.,20.beta.-diol | Pregnane-3,20-diol |

### 3.6. Role of Lactic Acid Dehydrogenate in the Fermented Leaves of C. dregei

LDH activity significantly increased ($p < 0.05$) with fermentation in a time-dependent manner in both the *T. fruiticosum* and the *C. dregei*. The LDH activity increased significantly in *C. dregei* compared to that of the *T. fruiticosum* after 3 days of fermentation (Figure 2). This illustrates the role of the lactic acid dehydrogenase enzyme as a preferred enzyme for *Enterococcus faecalis* fermentation of *C. dregei* leaves.

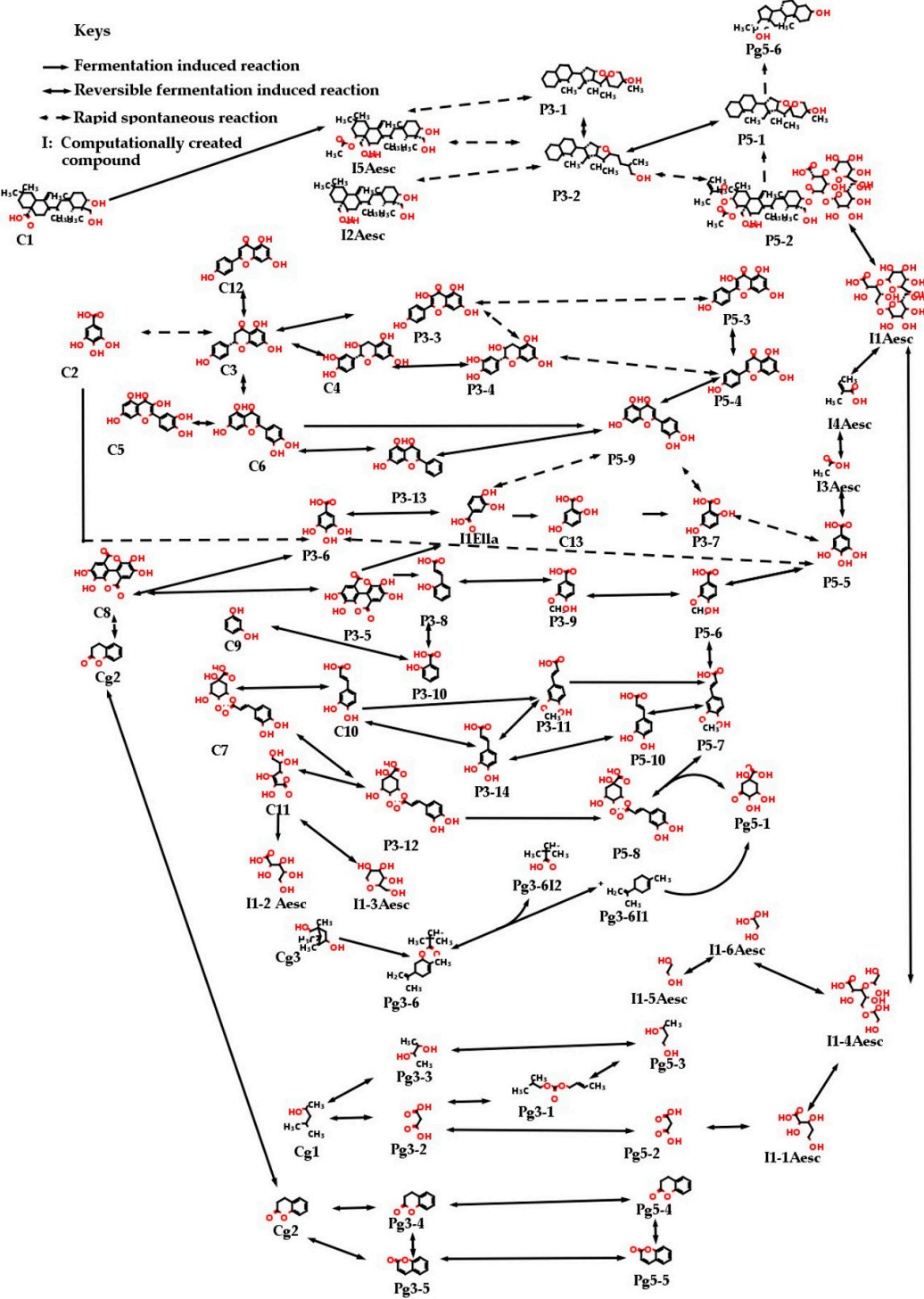

**Figure 1.** The proposed phytochemical transformation in the *E. faecalis*-induced fermented leaves of *C. dregei*. Key: **P3-1**: Spirostanol (0.01 mg/10 g); **P3-2: Furostanol** (47.86 mg/10 g); **P3-3**: Kaempferol

(2.74 mg/10 g); **P3-4**: Catechin (0.02 mg/10 g); **P3-5**: Ellagic acid (6.04 mg/10 g); **P3-6**: Gallic acid (29.81 mg/10 g); **P3-7**: 2,5-dihydroxybenzoic acid (0.31 mg/10 g); **P3-8**: O-coumaric acid (2.68 mg/10 g); **P3-9**: Vallinic acid (44.60 mg/10 g); **P3-10**: Salicylic acid (19.48 mg/10 g); **P3-11**: Ferulic acid (56.13 mg/10 g); **P3-12**: Chlorogenic acid (17.62 mg/10 g); P3-13: Chrysin (0.14 mg/10 g); P3-14: Caffeic acid (15.19 mg/10 g). **P5-1**: Spirostanol (60.24 %); **P5-2**: Aescin (39.72 mg/10 g); **P5-3**: Kaempferol (3.53 mg/10 g); **P5-4**: Apigenin (31.80 mg/10 g); **P5-5**: Gallic acid (10.44 mg/10 g); **P5-6**: Vallinic acid (11.65 mg/10 g); **P5-7**: Ferulic acid (48.35 mg/10 g); **P5-8**: Chlorogenic acid (11.76 mg/10 g); P5-9: Luteolin (20.54 mg/10 g); P5-10: Caffeic acid (61.705 mg/10 g); **Pg3-1**: (E)-But-2-enyl isobutyl carbonate (4.4%); **Pg3-2**: Malonic acid (4.53%); **Pg3-3**: 2,3-Butanediol (13.68%); **Pg3-4**: Hydrocoumarin (11.30%); **Pg3-5**: Coumarin (63.28%); **Pg3-6**: Limonen-6-ol, pivalate (1.62%); **Pg5-1**: 1,3,4-Trihydroxy-5-oxocyclohexanecarboxylic acid (1.42%); **Pg5-2**: Malonic acid (4.94%); **Pg5-3**: 1,3-Butanediol (9.30%); **Pg5-4**: Hydrocoumarin (18.96%); **Pg5-5**: Coumarin (64.13%); **Pg5-6**: Pregnane-3,20-diol (0.70%); **C1**: Hyderagenin (99.77 mg/10 g); **C2**: Gallic acid (44.83 mg/10 g); **C3**: Naringenin (6.74 mg/10 g); **C4**: Catechin (0.01 mg/10 g); **C5**: Quercetin (0.87 mg/10 g); **C6**:Luteolin (3.97 mg/10 g); **C7**: Chlorogenic acid (1.27 mg/10 g); **C8**: Ellagic acid (19.11 mg/10 g); **C9**: Resorcinol (0.21 mg/10 g); **C10**: Caffeic acid (21.75 mg/10 g); **C11**: Ascorbic acid (96.26 mg/10 g); C12: Apigenin (0.03 mg/10g); C13: 2,5-Dihydroxybenzoic acid (0.26 mg/10 g); **Cg1**: Isobutyl-methylmethanol (2.52%); **Cg2**: Hydrocoumarin (22.65%); **Cg3**: 1,7,7-Trimethylbicyclo[2.2.1]heptane-2,5-diol (53.34%); **I1Aesc**: (2S,3S,4R)-2,4-Dihydroxy-5-{[(2R,3R,4S,5S,6R)-3,4,5-trihydroxy-6-(hydroxymethyl)oxan-2-yl]oxy}-3-{[(2S,3R,4S,5S,6R)-3,4,5-trihydroxy-6-(hydroxymethyl)oxan-2-yl]oxy}pentanoic acid; **I2Aesc**: (3S,4S,4aR,6aR,6bS,8R,8aS,12aS,14aR,14bR)-4,8a-bis(hydroxymethyl)-4,6a,6b,11,11,14b-hexamethyl-1,2,3,4,4a,5,6,6a,6b,7,8,8a,9,10,11,12,12a,14,14a,14b-icosahydropicene-3,8-diol; **I3Aesc**: Acetic acid; **I4Aesc**: (2Z)-2-Methylbut-2-enoic acid; **I5Aesc**: (4S,4aR,5R,6aS,6bR,8aR,9S,10S,12aR,12bR,14bS)-5,10-Dihydroxy-4a,9-bis(hydroxymethyl)-2,2,6a,6b,9,12a-hexamethyl-1,2,3,4,4a,5,6,6a,6b,7,8,8a,9,10,11,12,12a,12b,13,14b-icosahydropicen-4-yl acetate; **I1-1Aesc**: (2S,3S)-2,3,5-trihydroxypentanoic acid; **I1-2Aesc**: Lyxonic acid; **I1-3Aesc**: D-Xylopyranose; **I1-4Aesc**: (2S,3S,4R)-3,5-bis[(1S)-1,2-dihydroxyethoxy]-2,4-dihydroxypentanoic acid; **I1-5Aesc**: Ethylene glycol; **I1-6Aesc**: Ethane-1,1,2-triol; **I1Ella**: 3,4-Dihydroxybenzoic acid; **Pg3-6I1**: 1-Methyl-4-(prop-1-en-2-yl)cyclohex-1-ene; **Pg3-6I2**: 2,2-Dimethylpropanoic acid.

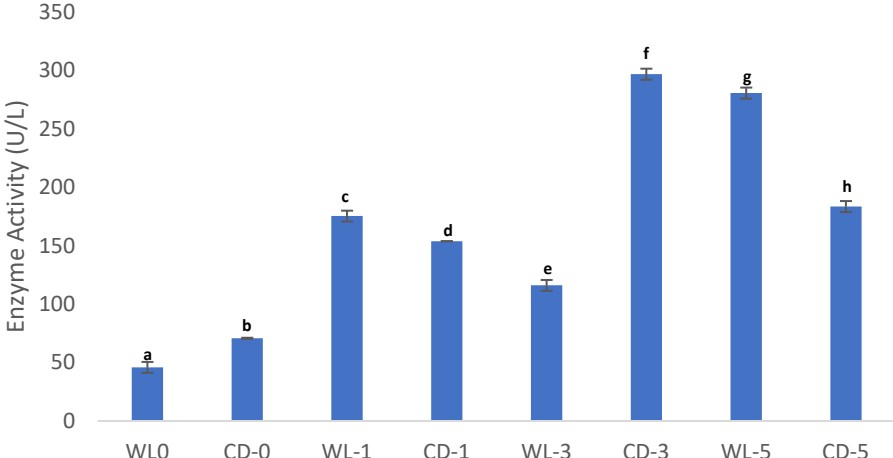

**Figure 2.** The activity of lactate dehydrogenase in fermented leaves of *C. dregei*. Results = mean $\pm$ SD ($n$ = 3). [a–h] The results with different superscript letters are significantly different ($p < 0.05$).

## 4. Discussion

The probiotic lactic acid bacteria (*E. faecalis*) adopted to propel the fermentation in this study was naturally non-virulent and considered safe since it has catalase-negative, non-hemolytic, and Gram-positive characteristics [11,22,23]. The organism facilitated the fermentation under alkaline conditions (Table 1), indicating that there were stronger alkaline compounds such as ammonia and hydroxyl compounds produced during the

fermentation process. Such fermentation was applied to improve nutrients and foods with health benefits and is usually known as alkaline fermentation [24–27].

*4.1. Nutritional Constituents in the Fermented C. dregei Leaves*

Moisture content, ash, and fat contents of both the fermented leaves of *T. fruticosum* and *C. dregei* increased throughout the fermentation period (Table 1). *C. dregei* leaves contain higher crude fiber than the normally consumed *T. fruticosum* without fermentation. These findings show that the leaves of *C. dregei* can help in keeping the digestive system functional and healthy [28]. However, the crude fiber content of the leaves of *C. dregei* increased as the fermentation progressed. The ash content followed a similar trend with an increased duration of fermentation. Ash content constituted the total mineral content in foods, and these minerals play a beneficial role in forming a nutritional physiochemical part of food [29]. The protein content of unfermented *C. dregei* is higher than *T. fruticosum* and increases over the fermentation days due to anabolism causing the proliferation of microbial cells and build-up of polymers, and fermentation could increase the free amino acids and nitrogen present [30]. The carbohydrate content of unfermented *C. dregei* is higher than that of the natural leaves of *T. fruticosum*. The fermentation process reduced the carbohydrate levels of the leaves of *C. dregei* below the value for *T. fruticosum*. This reduction in carbohydrates by fermentation might be because of the use of sugars as a source of carbon by organisms for metabolic activities. Thus, the reduction in these carbohydrate contents may be due to the respiratory activities of enzymes [31]. The moisture content is an important factor in fermentation as a result of secretion and biosynthesis of metabolites and microbial growth. The moisture content is lower in *C. dregei* than in normal leaves of *T. fruticosum*. The reduction decreases as fermentation progresses in *C. dregei*. This low moisture content in the fermented leaves of *C. dregei* indicates that it has a low water retention ability. High water activity can lead to lower substrate porosity, causing a reduction in gaseous exchange. The reverse (low water activity) may also lead to a decrease in microbial growth [32]. The fat content of unfermented *C. dregei* is lower than that of unfermented *T. fruticosum*. This fat content increases as fermentation time progresses. However, the levels were not as high as those of the fermented *T. fruticosum.* These findings could be due to a vast degradation of huge fatty molecules to simpler free fatty acid units because of the increased activity of lipolytic enzymes during fermentation [33].

The pattern of metabolic activities and physiological changes is strongly influenced by the availability of micronutrients in plants, animals, and humans. Micronutrient deficiencies result in malnutrition-related diseases such as kwashiorkor and anemia [34,35]. The leaves of *C. dregei* naturally possess superior values of calcium, selenium, magnesium potassium, sodium, zinc, and vitamin C, which are vital micronutrients for sustaining the normal metabolic processes in the body [34]. The specially endowed level of iron (14.94–18.75 mg/L) as compared to the 8.73 mg/L found in the commonly consumed water leaf is noteworthy. The leaves of *C. dregei*, however, possess lower values of vitamins A and E, which are strong antioxidants in nature (Table 2). The superior antioxidant values found in this study may therefore be due to the superior qualities of phytochemicals detected in the leaves of *C. dregei* (Tables 4 and 5).

The mineral levels in the leaves of *C. dregei* are comparable to those of regularly consumed water-leaf vegetables. The underutilized leaves of *C. dregei* possess adequate basic mineral nutrients that can meet the survival needs of potential consumers as indicated by this probiotic-facilitated fermentation. Notably, there was a gradual depletion of the sodium content of *C. dregei* in a time-dependent manner during the fermentation, suggesting it was a required substrate to facilitate the active biochemical process of *E. faecalis*-induced fermentation (Table 2). The fermentation expectedly depleted the levels of relatively harmful heavy metals (copper, iron, manganese, and cadmium) considered in this study. The leaves and the fermented leaves of *C. dregei* were rich in mineral iron, with the potential to easily provide the recommended daily intake for the mineral. The plant may be essential for enriching blood and treating anemia, sickle cell disease, and

other anemia-related diseases [36]. These leaves from *C. dregei* were also free of highly toxic elemental lead, indicating that there may be no heavy metal toxicity threat associated with their consumption [37,38]. From the antioxidant vitamin perspective, the probiotic-aided fermentation enhanced the water-soluble vitamin C concentration while depleting the values of fat-soluble vitamins A and E in the fermented leaves of *C. dregei*. The depreciation in the levels of vitamins A and E in the *C. dregei* leaves suggests the preference of the *E. faecalis*-fermenting organism for these fat-soluble vitamins during the fermentation (Table 2).

*4.2. Influence of Fermentation on the Antioxidants Qualities of C. dregei*

Plant nutrients are known for their countless health benefits to humans. These benefits include antioxidant, hormonal action, anti-microbial, anti-tumor, and anti-inflammatory. These bioactive chemicals neutralize the free radicals, as antioxidants. Thus, they protect DNA from damage and enhance carcinogen-detoxifying metabolites [39]. It is, therefore, possible that a strong antioxidant compound(s) was synthesized by the lactic acid bacteria, especially during the 5-day fermentation. The fermentation influences the phytochemical constituents present in these leaves between 1–5 days. The total concentrations of phytochemical compounds such as saponins, phenols, and flavonoid contents increased with the number of days of fermentation (Table 3). The physiological function of a plant on the body is determined by the phytochemical composition of the plant [40]. These compounds tend to be responsible for numerous health benefits. A typical example is saponins, which are metabolites with great importance in the beauty, drug, and food industries and are enhancers of the body's immune system as well as having certain cytotoxic properties [41]. Plant phenols aid the protection against pathogens, which thus helps in the treatment of pathogens, and their polyphenol compounds contain numerous health benefits, such as in blood pressure and glucose regulation, and exhibit anti-inflammatory, anti-viral, and anti-cytotoxic properties [42,43]. Metabolites such as terpenoids function as growth modulators in plants and in glycosylation in proteins. The carotenoids function as photosynthetic pigments, and the phytosterol serves as a component in membrane dynamics in humans. Saponin also serves as an anti-malarial and anti-diuretic in pharmaceutical drug formulations [39].

*4.3. Phytochemical Constituents in the Leaves of the Fermented C. dregei*

The phytochemical configuration of plants could reveal their physiological functions in the body and their health benefits [40]. The fermentation improved the phenolic makeup of compounds in the fermented leaves (Table 4). The fermentation aided the synthesis of spirostanol (60.24 mg/10 g extract), furostanol (47.86 mg/10 g extract), aescin (39.72 mg/10 g extract), ferulic acid (56.13 mg/10 g extract), chlorogenic acid (5.04 mg/10 g extract), kaempferol (3.53 mg/10 g extract), chrysin (0.14 mg/10 g extract), caffeic acid (61.71 mg/10 g extract), o-coumaric acid (2.68 mg/10 g extract), salicylic acid (19.48 mg/10 g extract), vanillic acid (44.60 mg/10 g extract), apigenin (26.47 mg/10 g extract), and two freshly discovered saponin (P-Scd, 52.05 mg/10 g extract), and phenolic (P-Pcd, 0.23 mg/10 g extract). One unidentified phenolic compound (S-Pcd) and one bioflavonoid compound (S-Bcd) naturally occurred in the leaves of *C. dregei*, and another unidentified newly synthesized phenolic compound (P-Pcd) was facilitated by *E. faecalis*-induced fermentation (Table 4). Coumarin (63.28–64.13%) was the predominant volatile compound synthesized by the 5-day *E. faecalis*-induced fermentation. The presence of hydrocoumarin in the natural leaves of *C. dregei* is a possible precursor for the synthesis of the coumarin compound. Thus, *C. dregei* leaves are a good source of gallic acid which is usually found in vegetables such as onion and garlic that were recently linked to combat SARS-Cov2 virus and hence cure for Covid-19 pandemic disease [44–46]. Gallic acid was similarly reported in the fermented leaves of *Solenostemon monostachyus* [3]. The consumption of vegetables with good sources of gallic acid serves to detoxify the toxic effects due to some phytochemicals and possesses therapeutic potential [44]. Fortunately, the health-beneficial compound was sustained

in the fermented leaves of *C. dregei* to continue this antiviral role. Of all the twenty-one (21) phytochemical compounds detected as naturally occurring in leaves of *C. dregei*, only resorcinol, though present in small amounts, was found to have mild toxicity [47]. This fermentation process eliminated the resorcinol toxicity concern, which is expected since fermentation was reported to improve the safety of foods [48]. This study also revealed chlorogenic acid can be biosynthesized from ferulic acid and 1,3,4-Trihydroxy-5-oxocyclohexanecarboxylic acid (Pg5-1), and vice versa. Aescin seems to be the most complex phytochemical compound produced in the fermented leaves of *C. dregei* (Table 4). This pattern of biosynthesizing aescin is a total reverse form reported in the fermented leaves of *S. monostachyus* where aescin was the precursor molecule from which other simpler phytochemical compounds were derived [3]. Ellagic acid is a highly sought-after phytochemical due to its associated health benefits such as anti-mutagenic, anti-diabetic, anti-inflammatory, and anti-carcinogen [49]. It is a major bioactive in tea, red wines, walnut, and several fruit and vegetables most especially when they are fermented [50,51]. Ellagic acid was successfully linked to improving postharvest qualities and the antioxidant potential of fruits [52].

Extending the fermentation period to 5 days produced two entirely new compounds (spirostanol and aescin). Interestingly, one of these two saponins, spirostanol, was the product of fermenting another plant (*S. monostachyus*) with *E. faecalis* for the same duration [3]. Spirostanol and furostanol are among the major phytochemical compounds in the *E. faecalis*-aided fermented leaves of *C. dregei*. The two compounds are commonly linked to fermented food products. The biosynthesis of furostanol from spirostanol was previously reported [53,54]. Formic and gallic acids have been previously linked to fermentation as reported in this study [55,56].

Cg4 (1,7,7-trimethylbicyclo[2.2.1]heptane-2,5-diol) is a major volatile compound (53.34%) that naturally occurs in the leaves of *C. dregei*. The compound was also previously linked to similar lactic acid bacterial-aided juice fermentation [57]. Fermentation remains a mystery to humans, and increasing efforts are being made to elucidate this process [58–62]. The application of fermentation is ancient and cuts across all continents. The pathway illustrated as a scheme in this study (Figure 1) is an advanced form of the phenylpropanoid pathway since it consists of several other molecules in addition to those molecules already elucidated in the phenylpropanoid pathway [63]. Pathways provide the required understanding to exploit the depth of knowledge associated with metabolic processes [64,65]. We believe that the biochemical scheme elucidated in this study should facilitate humans to better utilize fermentation to advance the application of nature to provide quality nutrients, bioactive phytochemicals, and food security.

Four novel intermediary compounds, (2S,3S,4R)-2,4-dihydroxy-5-{[(2R,3R,4S,5S,6R)-3,4,5-trihydroxy-6-(hydroxymethyl)oxan-2-yl]oxy}-3-{[(2S,3R,4S,5S,6R)-3,4,5-trihydroxy-6-(hydroxymethyl)oxan-2-yl]oxy}pentanoic acid (I1Aesc), (4S,4aR,5R,6aS,6bR,8aR,9S,10S,12aR,12bR,14bS)-5,10-dihydroxy-4a,9-bis(hydroxymethyl)-2,2,6a,6b,9,12a-hexamethyl-1,2,3,4,4a,5,6,6a,6b,7,8,8a,9,10,11,12,12a,12b,13,14b-icosahydropicen-4-yl acetate (I5Aesc), (2S,3S)-2,3,5-trihydroxypentanoic acid (I1-1IAesc), and (2S,3S,4R)-3,5-bis[(1S)-1,2-dihydroxyethoxy]-2,4-dihydroxypentanoic acid (I1-4Aesc), were among the fermentation-linked *E. faecalis*-induced compounds computationally created in this study (Figure 1). Six other established compounds, I2Aesc, I1-1IAesc, I1-2Aesc, I1-3Aesc, I1-5Aesc identified as ethane-1,2-diol (or ethylene glycol), and I1-6Aesc identified as ethane-1,1,2-triol (oxyethylene glycol), were computationally designed and first reported as metabolites of fermentation in this study (Figure 1).

I1-4Aesc is probably derived from 2,3-dihydroxypentanoic acid, while I1-1IAesc has a typical basic glucose structural skeleton with a few missing hydroxyl groups. This novel compound is probably an isomer of 2,3,4-trihydropentanoic acid and 2,4,5-trihydropentanoic acid that mainly exists in ester forms [66,67]. I1-2Aesc was identified as lyxonic acid and was previously reported in *Vitis vinifera* (wine grape). This compound is one of the end products of ascorbic acid degradation recently reported [68,69]. I1-3Aesc is a D-Xylopyranose

compound that is similar to the aldopentoses (ribose, arabinose, xylose, and lylose) and can be described as one unit of amylotriose with one less hydroxyl group. We speculate the possible existence of amylotriose dehydratase enzyme in the fermenting organism that can metabolize the removal of hydroxyl groups to produce I1-3Aesc [70–73]. I1-2Aesc was proposed to be rapidly biotransformed into ascorbic acid in this study, probably due to the ability of the lactic acid bacteria (*E. faecalis*) to biosynthesize the enzyme to facilitate the rare removal of hydroxyl groups in biological systems [74]. It is known that pentose compounds are usually transformed into furan structures as found in amino acids when dehydrated [75].

Aescin was identified as a product of fermenting the leaves of *C. dregei*, which is contrary to the findings in a similar report where it was eliminated in another fermentation study on the leaves of *S. monostachyus* [3]. Vanillic acid, salicylic acid, ellagic acid, coumarin, quercetin, catechin, and resorcinol were among the important phytochemicals in the *C. dregei* fermentation study that differed from the phytochemical compounds reported in a similar *E. faecalis*-fermentation-aided study on the leaves of *S. monostachyus* (Table 4). Fermentation aided the destruction of resorcinol, which is present in the raw leaves of *C. dregei*. Resorcinol is the key strong antioxidant bioactive in a Japanese tea produced by anaerobic fermentation aided by *E. faecalis* and is sometimes used to control enzyme browning in food such as crustaceans if carefully controlled levels are maintained to avoid toxicity [47,76]. The associated toxicity of resorcinol may not be of concern in the consumption of the fermented leaves of *C. dregei,* so there may be no harm when consumed in large quantities. The production of acid and alcohol is typical in most fermentation processes. Malonic acid (4.53–4.94%) is a major fatty acid produced, while dimethylethylene glycol (13.68–18.96%) appears to be the major alcohol generated by the 3–5-day fermentation of the leaves of *C. dregei* (Tables 4 and 5). The presence of malonic acid may indicate the involvement of tricarboxylic acid cycle biochemical processes in generating energy for fermentation [77,78]. The high level of dimethylethylene glycol compared to the level of malonic acid produced may be the major factor influencing the continual increase in the alkaline level (3.98–8.50–8.90) of the fermenting medium in this study (Table 1). The compound (I4-AeSC) is novel and uniquely implicated in this fermentation study (Figure 1). Ascorbic acid and chlorogenic acid are often present together in foods such as mushrooms and vegetables. Both compounds were also implicated in the browning of food crops [79–83]. The fermentation scheme in this study suggests biosynthesizing chlorogenic acid from ascorbic acid and vice versa under *E. faecalis*-induced fermentation conditions as a possible reason for the conjugal presence in food (Figure 1). Ascorbic acid was established as a biosynthetic precursor of most acids (tartaric acid, oxalic acid, 2-keto-L-idonic acid, L-idonic acid, gluconic acid, tetruronic acid, dehydroascorbic acid, and L-threonic acid) in plants [84]. This study further reveals ascorbic acid as a precursor of chlorogenic acid biosynthesis in plants, which explains why both compounds are often present together in plant foods.

It was difficult to identify the role of ethyl 5,8,11,14,17-icosapentaenoate (Et-conoate) in the scheme since the chemistry of metabolism of the compound has not been studied in detail, although it has already been identified [85]. It may, however, serve as the main aroma-providing compound for fermented food products [86]. The ester of arachidonic acid was catabolized by the 3–5-day *E. faecalis*-induced fermentation (Table 5). The product(s) of microbial degradation and the associated biochemical process is yet to be elucidated. However, the compound is an ester of arachidonic acid, and the account of microbial synthesis of the arachidonic acid through combined efforts of elongase and desaturase enzymes was reported with an extensive metabolic pathway [87–90]. The general catabolism of long unsaturated fatty acids such as arachidonic acid occurs in the peroxisomes following the combined action of dehydrogenation, hydration, dehydration, and thiolic cleavage processes. The first process is linked to the biochemical action of acyl-CoA oxidase 1, facilitating the conversion of acyl-CoA to trans-2-enoyl-CoA. The next two processes were both

facilitated by the same enzyme, 17β-hydroxysteroid dehydrogenase 4, while the fourth process was actioned by the peroxisomal 3-oxoacyl-CoA thiolase [85].

It is on this premise that we proposed that the Et-connate microbial catabolism could be fractioned into a combination of three (3) molecules of but-1-ene and one molecule of (2E)-but-2-ene and ethylbutanoate or fractions of two (2) molecules of (2E)-pentene, one molecule of penta-1,4-diene, and ethyl pentanoate. We suggest that the latter fraction is most appropriate since many pentanoate compounds (such as 5-hydroperoxyeicosatetraenoic acid) are associated with arachidonic acid metabolism [89]. 5-Hydroperoxyeicosatetraenoic acid or dihydroxyeicosatrienoic acids and their structural analogs were linked as derivatives of arachidonic acid catabolism when facilitated by the lipoxygenases and W-hydroxylase enzymes [89,91]. This provides an understanding that implicates the four possible compounds, I4-Aesc, I5Aesc, I1-4Aesc, and Pg3-6I2, identified in our scheme as products of the metabolism of Et-conoate, although we envisaged this as mainly facilitated by the *E-faecalis*-induced fermenting process. Fatty acid serves as a precursor of phytochemicals as reported in the conversion of plasma membrane fatty acids to myristic acid [92]. It is known that different elongases exhibit diverse preferences in the number of carbons, number of hydroxyl groups, and degree of unsaturation. A typical elongase with such unique preferences was implicated in the biosynthesis of erucic acid in jojoba (*Simmondsia chinensis*), *Trypanosomiasis brucei*, and *Saccharomyces cerevisiae* [92]. B-oxidation was established to cleave two carbons at every cycle and stop when a 4-carbon or 5-carbon compound remains, as in the case of fatty acids with even- or odd-numbered chain length, respectively [85]. Other types of oxidation of fatty acids exist that cleave a 4-carbon compound per cycle, as identified in catfish equipped with unique enzymes [93].

The arachidonic acid metabolic pathways suggest a possible conversion of Et-conoate to a cyclic form when facilitated by cyclooxygenase [89], which strongly implicates the (1S,4S)-bornane-2alpha,5beta-diol (Cg3) detected in the natural leaves of *C. dregei*.

### 4.4. Influence of Lactic Acid Dehydrogenase on the Fermentation of the Leaves of C. dregei

Lactic acid dehydrogenase activity (LDH) in both the fermented *T. fruticosum* and *C. dregei* leaves increases throughout the fermentation period [94]. The LDH activity was higher than that recorded for the unfermented *T. fruticosum* after 3 days of fermentation. This trend was highly conspicuous on the third day of fermentation of *C. dregei* leaves. Lactate dehydrogenase enzyme is an enzyme that catalyzes the conversion of pyruvate or pyruvic acid into lactic acid, which occurs mainly during anaerobic lactate fermentation. Increased activity of lactate dehydrogenase during fermentation proves that there is constant catabolism of sugars into pyruvate in cellular materials. Despite this, human beings do not possess D-lactic acid dehydrogenase, and therefore this enzyme cannot be consumed. L-lactate is mainly metabolized by humans. As a result, L-lactate is utilized in food and pharmaceutical industries. Optically viable L-lactic acid can only be produced through microbial and enzyme fermentation [95]. *Enterococcus faecalis* is a probiotic lactic acid bacterium linked to lactic acid dehydrogenase, which influences most anaerobic fermentation. In this study, the lactic acid dehydrogenase was conveniently utilized during the fermentation of the leaves of *C. dregei*, which is quite contradictory to the preference for alpha-amylase in a report on fermentation of *Solenostemon monostachyus* leaves aided by the same probiotic organisms [3]. The factors that influence enzyme preference in fermenting organisms are yet to be clearly understood. However, the choice of such an enzyme for fermentation may be due to the characteristics of the phytochemicals presented as a substrate for the fermenting organisms.

## 5. Conclusions

The leaves and the fermented leaves of *C. dregei* were rich in mineral iron, with the potential to easily provide the recommended daily intake for the mineral. The plant may be essential for enriching blood and treating anemia, sickle cell disease, and other anemia-related diseases. These leaves from *C. dregei* were also free of highly toxic elemental

lead, indicating that there may be no heavy metal toxicity threat associated with their consumption. Four novel intermediary compounds and six other established compounds were freshly identified with fermentation. *C. dregei* is rich in nutrients and has strong antioxidant qualities, which were improved by the fermentation technique. *E. faecalis* is most likely to engage LDH in driving the fermentation transforming the *C. dregei* into a potential edible vegetable. Fermentation has the potential to improve the nutritional and health properties of vegetable products by enhancing their nutrients and phytochemical constituents. The present war against worldwide hunger may be addressed by utilizing the leaves of *C. dregei* as a supplementary source of vegetables for humans. More underutilized leaves or seeds from plants can be transformed into edible food and a source of health-beneficial phytochemicals using this fermentation technique. The fermentation technique may also apply to transforming the leaves from *C. dregei* into wines, which may benefit the wine-producing industries. It is also possible to utilize the fermenting organism as a useful tool to biosynthesize specific bioactive compounds from a toxic compound and also bioremediate heavy metals in polluted water or rivers.

**Supplementary Materials:** The following supporting information can be sourced from: https://www.mdpi.com/article/10.3390/fermentation9080707/s1, Figure S1: HPLC chromatogram for identification of vitamin C standard; Figure S2: HPLC chromatogram for identification of vitamin C in the unfermented *T. fruticosum* leaves; Figure S3: HPLC chromatogram for identification of vitamin C in the unfermented *C. dregei* leaves; Figure S4: HPLC chromatogram for identification of vitamin C in the 3-day-fermented *C. dregei* leaves; Figure S5: HPLC chromatogram for identification of vitamin C in the 5-day-fermented *C. dregei* leaves; Figure S6: HPLC chromatogram for identification of vitamin A standard; Figure S7: HPLC chromatogram for identification of vitamin E standard; Figure S8: HPLC chromatogram for identification of vitamin A and E in the unfermented *T. fruticosum* leaves; Figure S9: HPLC chromatogram for identification of vitamin A and E in the unfermented *C. dregei* leaves; Figure S10: HPLC chromatogram for identification of vitamin A and E in the 3-day-fermented *C. dregei* leaves; Figure S11: HPLC chromatogram for identification of vitamin A and E in the 5-day-fermented *C. dregei* leaves; Figure S12: HPLC chromatogram for identification of saponin compounds in the unfermented *C. dregei* leaves; Figure S13: HPLC chromatogram for identification of saponin compounds in the 3-day-fermented *C. dregei* leaves; Figure S14: HPLC chromatogram for identification of saponin compounds in the 5-day-fermented *C. dregei* leaves; Figure S15: HPLC chromatogram for identification of phenolic compounds in the unfermented *C. dregei* leaves; Figure S16: HPLC chromatogram for identification of phenolic compounds in the 3-day-fermented *C. dregei* leaves; Figure S17: HPLC chromatogram for identification of phenolic compounds in the 5-day-fermented *C. dregei* leaves; Figure S18: HPLC chromatogram for identification of bioflavonoid compounds in the unfermented *C. dregei* leaves; Figure S19: HPLC chromatogram for identification of bioflavonoid compounds in the 3-day-fermented *C. dregei* leaves; Figure S20: HPLC chromatogram for identification of bioflavonoid compounds in the 5-day-fermented *C. dregei* leaves; Figure S21: GC/MS chromatogram for identification of volatile phytochemicals in the unfermented *C. dregei* leaves; Figure S22: GC/MS chromatogram for identification of volatile phytochemicals in the 3-day-fermented *C. dregei* leaves; Figure S23: GC/MS chromatogram for identification of volatile phytochemicals in the 5-day-fermented *C. dregei* leaves.

**Author Contributions:** Conceptualization, I.S.A. and O.E.O.; Methodology, I.S.A., A.J.A., P.A.G., E.F.A. and A.O.A.; Formal analysis, I.S.A., A.J.A. and P.A.G.; Investigation, I.S.A., A.J.A., P.A.G., E.F.A. and A.O.A.; Resources, O.E.O.; Data curation, I.S.A.; Writing—original draft, I.S.A. and A.J.A.; Writing—review and editing, I.S.A. and E.F.A.; Visualization, I.S.A.; Supervision, I.S.A.; Project administration, I.S.A. All authors have read and agreed to the published version of the manuscript.

**Funding:** The authors declare no external funding in support of this research study. The expected publication support from Covenant University management funding the article processing fees of this article is highly appreciated.

**Data Availability Statement:** The detailed sequence and the accession number (MW481698) of the fermenting organism (*E. faecalis*) for this study are available in the National Centre for Biotechnology Information (NCBI) database—https://www.ncbi.nlm.nih.gov/nuccore/MW481698 (accessed on 18 November 2022). The fermenting organism can also be made available on request.

**Acknowledgments:** The authors appreciate the approval and award of a license permitting the use of software tools developed by ChemAxon Ltd., Budapest, Hungary, for the chemical drawings and computational prediction of the reaction scheme in this study.

**Conflicts of Interest:** The authors have no conflicts of interest associated with this study.

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
