# Peer review of "Enterococcus faecalis-Aided Fermentation to Facilitate Edible Properties and Bioactive Transformation of Underutilized Cyathea dregei Leaves"

_fermentation, doi:10.3390/fermentation9080707_

Round 1
Reviewer 1 Report
Comments to authors
Comments to authors
Fermentation- 2454684
In the research, Enterococcus faecalis-Aided Fermentation to Facilitate Edible Properties and Bioactive Transformation of Underutilised Cyathea dregei Leaves, scientifically interesting data is available. The results and the discussion parts are well written, authors have elaborated on all the points. I recommend this article for publication after improving some comments and questions listed below.
The language of the manuscript should be revised by a native speaker very carefully. The repetition of words is present, the author should carefully read the manuscript and avoid repetition. For example Line, 18.
1. The abstract needs a drastic revision. There are some grammatical mistakes in the abstract and in the manuscript that could be hard to understand by readers. For example lines 21-24. Please correct the phrase from Line 29, ‘’C. dregei is nutritionally rich’’.
2. The introduction part needs some improvements. There is very small data on the importance and need for this study. Moreover, the author should check the link of paragraphs, there is no link between them.
3. Section 2.2, rewrite and use correct prepositions.
4. Lines 135-136, are grammatically incorrect.
5. Lines 206-208, are confusing, please rewrite them.
6. Lines 235-238, the phrase is vague and difficult to understand.
7. The author should read the whole manuscript carefully to solve the grammatical problems and long sentences should be avoided for better understanding of readers.
The language of the manuscript should be revised by a native speaker very carefully. The repetition of words is present, the author should carefully read the manuscript and avoid repetition.
Author Response
Dear Reviewer,
We have attended to the comments raised during the review of our manuscript. The comments were indeed beneficial, constructive, and have immensely improved the quality and understanding of the manuscript. Authors appreciate the efficient and fast review process. We have uploaded the word document (with the trace of the changes effected), and the PDF version of the revised manuscript as requested for your kind consideration. Kindly find below our point-by-point response to your comments.
Reviewer 1
Comment 1-1. The abstract needs a drastic revision. There are some grammatical mistakes in the abstract and in the manuscript that could be hard to understand by readers. For example lines 21-24. Please correct the phrase from Line 29, ‘’C. dregei is nutritionally rich’’.
Response: The sentences indicated had been revised as suggested to improve meaning.
Comment 1-2. The introduction part needs some improvements. There is very small data on the importance and need for this study. Moreover, the author should check the link of paragraphs, there is no link between them.
Response: Lines 59-63 (in the last manuscript) provide data supporting food insecurity and population growth that threatens to worsen the situation. We provided further information on the implication of malnutrition on the health of humans. Both the old and the new information were harmonized and repositioned to avoid repetition (See lines 42-54 in the revised manuscript).
Comment 1-3. Section 2.2, rewrite and use correct prepositions.
Response: We critically revised the section as suggested.
Comment 1-4. Lines 135-136, are grammatically incorrect.
Response: The grammatical errors have been corrected as suggested.
Comment 1-5. Lines 206-208, are confusing, please rewrite them.
Response: The statement has been revised to improve clarity as suggested.
Comment 1-6. Lines 235-238, the phrase is vague and difficult to understand.
Response: The statement has been revised accordingly to improve understanding.
Comment 1-7. The author should read the whole manuscript carefully to solve the grammatical problems and long sentences should be avoided for better understanding of readers.
Response: The vague sentences have been revised and long sentences simplified as suggested.
Reviewer 2 Report
The topic of the manuscript (fermentation-2454684) is of interest because the natural product is constantly growing. The manuscript describes in general an interesting laboratory work, which could be useful for other researchers and/or industry working on the topic. In the study, the impact of Enterococcus faecalis as an agent of 3-5 days-fermentation was investigated. The proximate content, biochemical, antioxidant properties, and phytochemical constituents were examined on the unfermented and fermented leaves. The results indicated that E. faecalis is most likely to engage LDH in driving the fermentation transforming the C. dregei into possible vegetable food. Regarding results, some concepts should be carefully revised. There are many problems that have not been explained clearly.
1. In the figure 2, significance needs to be marked on the bar chart.
2. The number of significant digits should be consistent in the text.
Author Response
Dear Reviewer,
We have attended to the comments raised during the review of our manuscript. The comments were indeed beneficial, constructive, and have immensely improved the quality and understanding of the manuscript. Authors appreciate the efficient and fast review process. We have uploaded the word document (with the trace of the changes effected), and the PDF version of the revised manuscript as requested for your kind consideration. Kindly find below our point-by-point response to your comments.
Reviewer 2
Comments 1. In the figure 2, significance needs to be marked on the bar chart.
Response: All the bars in Figure 2 with significant differences have been identified with superscript letters as suggested.
Comments 2. The number of significant digits should be consistent in the text.
Response: The number of digits in the manuscript has been uniformly revised to two decimal points as suggested.
Reviewer 3 Report
I revised the manuscript in title (Enterococcus faecalis-Aided Fermentation to Facilitate Edible Properties and Bioactive Transformation of Underutilised Cyathea dregei Leaves) and it have an interesting information and written will. I have some suggestions to improve it.
- Line 27, Which phenolic is it total phenolic or phenolic compounds (in abstract we dont use abbreviations)
- Line 77, Afolabi et al., Revise the reference citation in the text like journal format
- Line 89, Authors should describe the methods in details
- Line 102, Add manufacture, cite, country
- Line 114, The methods used must described in the manuscript
- Line 129, which procedure it must described here
- Line 144, Write the extraction method in details
- Line 145, What are the HPLC conditions used
- Line 163, What are the GC-MS conditions used
- Line 168, Add the method of extraction and the determination protocol, manufacture, city and country of the Kit
- In table 1 and 2, authors used strange letters , What is this means
- Line 214-216, Move it in discussion part with reference
- Line 222, Which antioxidant vitamins
- In conclusion, Add sentence about the application of this work and future view in its use and who is the benefits

I revised the manuscript in title (Enterococcus faecalis-Aided Fermentation to Facilitate Edible Properties and Bioactive Transformation of Underutilised Cyathea dregei Leaves) and it have an interesting information and written will. I have some suggestions to improve it.
- Line 27, Which phenolic is it total phenolic or phenolic compounds (in abstract we dont use abbreviations)
- Line 77, Afolabi et al., Revise the reference citation in the text like journal format
- Line 89, Authors should describe the methods in details
- Line 102, Add manufacture, cite, country
- Line 114, The methods used must described in the manuscript
- Line 129, which procedure it must described here
- Line 144, Write the extraction method in details
- Line 145, What are the HPLC conditions used
- Line 163, What are the GC-MS conditions used
- Line 168, Add the method of extraction and the determination protocol, manufacture, city and country of the Kit
- In table 1 and 2, authors used strange letters , What is this means
- Line 214-216, Move it in discussion part with reference
- Line 222, Which antioxidant vitamins
- In conclusion, Add sentence about the application of this work and future view in its use and who is the benefits
Author Response
Dear Reviewer,
We have attended to the comments raised during the review of our manuscript. The comments were indeed beneficial, constructive, and have immensely improved the quality and understanding of the manuscript. Authors appreciate the efficient and fast review process. We have uploaded the word document (with the trace of the changes effected), and the PDF version of the revised manuscript as requested for your kind consideration. Kindly find below our point-by-point response to your comments.
Reviewer 3
Comment 1- Line 27, Which phenolic is it total phenolic or phenolic compounds (in abstract we dont use abbreviations).
Response: It is a phenolic compound. The statement has been revised accordingly.
Comment 2- Line 77, Afolabi et al., Revise the reference citation in the text like journal format.
Response: The previous reference style was retained as the current endnote style recommended by the Journal and the publisher was used. The link to the present endnote style indicated in the Instruction for Authors is herewith provided for verification: https://endnote.com/style_download/mdpi/.
Comment 3- Line 89, Authors should describe the methods in details.
Response: We have included the following detailed methods for the collection and preparation of fermentative Lactic Acid Bacteria as requested.
Comment 4- Line 102, Add manufacture, cite, country.
Response: The requested detail (GenLab OV/200/SS/F/DIG oven, Widnes WA8 0SR, UK) of the oven used has been incorporated into the revised manuscript.
Comment 5- Line 114, The methods used must described in the manuscript.
Response: The method for each parameter has been provided as suggested (See section 2.5.1.).
Comment 6- Line 129, which procedure it must described here.
Response: The method for each parameter has been provided as suggested (See section 2.5.2.).
Comment 7- Line 144, Write the extraction method in details.
Response: The word extraction as previously used is now qualified to indicate it was for extracted samples for better clarity. Also, the detailed preparation methods of the HPLC samples for the analysis of the vitamins have been incorporated into the revised manuscript for clarity.
Comment 8- Line 145, What are the HPLC conditions used.
Response: The HPLC conditions used have been incorporated as suggested.
Comment 9- Line 163, What are the GC-MS conditions used.
Response: The requested GC-MS conditions have been provided in the revised manuscript.
Comment 10- Line 168, Add the method of extraction and the determination protocol, manufacture, city and country of the Kit.
Response: The detail of the determination protocol and other requested detail of the kit has been provided. The manufacturer (Randox) was previously identified with the kit, while the protocol was revised to indicate that the extracts from the leaves were directly used as the source of the lactate dehydrogenase enzyme for the analysis (Please see line 201-208).
Comment 11- In table 1 and 2, authors used strange letters, What is this means?
Response: The symbol (* and ¥) have been replaced with superscript letters which were previously defined below the tables 1 and 2.
Comment 12- Line 214-216, Move it in discussion part with reference.
Response: The indicated statement (The plant may be essential for enriching blood and treating anemia, sickle cell, and other anemia-related diseases) was already discussed with reference (lines 427-429 in the last manuscript and lines 453-455 in the revised manuscript).
Comment 13- Line 222, Which antioxidant vitamins.
Response: The antioxidant vitamins refer to vitamins A, E, and C. These vitamins were indicated in the later part of the sentence (See line 261-263 in the revised manuscript and line 222-224 in the last manuscript).
Comment 14- In conclusion, add sentence about the application of this work and future view in its use and who is the benefits.
Response: We have incorporated the following statement to reflect the application of this study and the future perspective (See line 675-681 of the revised manuscript) – More underutilised leaves or seeds from plants can be midwives into edible food as a source of health-beneficial phytochemicals using this fermentation technique. The fermentation technique may also apply to transforming the leaves from C. dregei into wines, which may benefit the wine-producing industries. It is also possible to engage the fermenting organism as a useful tool to biosynthesise specific bioactive compounds from a toxic compound, and also bioremediate heavy metals in polluted water or rivers.